# Safety Alignment Shouldn't Be Complicated

**WARNING: This paper contains model-generated content that can be offensive in nature.**

## Abstract

As large language models (LLMs) are overwhelmingly more and more integrated into various applications, ensuring they generate safe and aligned responses is a pressing need. Previous research on alignment has largely focused on general instruction-following but has often overlooked the unique properties and challenges of safety alignment, such as the brittleness of safety mechanisms. To bridge the gap, we propose the Superficial Safety Alignment Hypothesis (SSAH), which posits that safety alignment should teach an otherwise unsafe model to choose the correct reasoning direction - interpreted as a specialized binary classification task - and incorporate a refusal mechanism with multiple reserved fallback options. Furthermore, through SSAH, we hypothesize that safety guardrails in LLMs can be established by just a small number of essential components. To verify this, we conduct an ablation study and successfully identify four types of attribute-critical components in safety-aligned LLMs: Exclusive Safety Unit (ESU), Exclusive Utility Unit (EUU), Complex Unit (CU), and Redundant Unit (RU). Our findings show that freezing certain safety-critical components **(7.5%)** during fine-tuning allows the model to retain its safety attributes while adapting to new tasks. Additionally, we show that leveraging redundant units **(20%)** in the pre-trained model as an "alignment budget" can effectively minimize the alignment tax while achieving the alignment goal. All considered, this paper concludes that the atomic functional unit for safety in LLMs is at the neuron level and underscores that safety alignment should not be complicated. We believe this work contributes to the foundation of efficient and scalable safety alignment for future LLMs.

## 1 Introduction

Large language models (LLMs) are demonstrating remarkable capabilities across a broad spectrum of natural language tasks, ranging from text generation to answering complex questions (Achiam et al., 2023; Touvron et al., 2023a;b; Dubey et al., 2024). However, as these models are increasingly integrated into real-world applications, concerns about the risk of generating harmful, unsafe, or unethical content have grown Askell et al. (2021); Bai et al. (2022); Zeng et al. (2024). This has led to a pressing need for safety alignment, which ensures that LLM outputs are not only coherent and informative but also aligned with human values, ethical standards, and safety considerations.

Previous research on alignment has primarily focused on enhancing LLMs' ability to follow general instructions without enough attention to model safety. This trend of treating safety alignment as a subset of general alignment has obscured its distinct challenges (Ouyang et al., 2022; Rafailov et al., 2024; Zhou et al., 2024; Liu et al., 2023a; Yuan et al., 2023; Liu et al., 2023b). One major issue is the brittleness of current safety mechanisms. Despite using benign data during model fine-tuning, (Qi et al., 2023; Yang et al., 2023) have shown that safety mechanisms can fall apart when models are adapted to new tasks. This failure mode amplifies the brittle nature of current safety alignment techniques. Furthermore, safety alignment also comes with an 'alignment tax' - a trade-off where improving safety may reduce the model's overall utility or downstream task performances Ouyang et al. (2022); Wang et al. (2024); Lin et al. (2024). Additionally, current safety alignment approaches

typically require full model fine-tuning to ensure the final performance Hu et al. (2021); Zhang et al. (2023); Thakkar et al. (2024), which is computationally costly. To address these issues, we must accurately understand *how safety alignment impacts model behavior, why safety mechanisms are brittle*, and *how to mitigate the alignment tax*.

Recently, Wei et al. (2024b) claimed the fragility of safety mechanisms in LLMs by identifying safety-critical components at the weight level through pruning (see Appendix C.2 for details). However, the work did not thoroughly address the fundamental issues discussed above nor propose effective solutions to resolve them. This paper introduces the **Superficial Safety Alignment Hypothesis (SSAH)**, which separates safety alignment from general alignment and emphasizes its unique properties. With this hypothesis, we address that *less is more* in the context of safety alignment - a small number of but strategically vital safety-critical components are sufficient to achieve robust safety performance. Our extensive experiments prove that by freezing certain safety-critical computational units, safety performance is successfully preserved against fine-tuning attacks, and we are the ***first*** to do it. Additionally, we introduce the concept of repurposing redundant units within the pre-trained model as an "alignment budget." By reassigning these units toward safety tasks, we can significantly reduce the alignment tax, ensuring that safety is maintained without sacrificing the model's overall utility. Taken all together, this paper addresses the following core questions:

**Question 1.** *How does safety alignment impact model behavior?*

**Answer:** Through **SSAH**, we posit that safety alignment fundamentally alters a model's decision-making process by teaching an otherwise unsafe model - fulfilling malicious or harmful requests - to follow the correct reasoning pathways. This process can be viewed as a specialized binary classification task - the model must either fulfill the user's request or refuse it based on safety considerations. Additionally, safety alignment equips the model with a standard refusal mechanism, along with reserved fallback options.

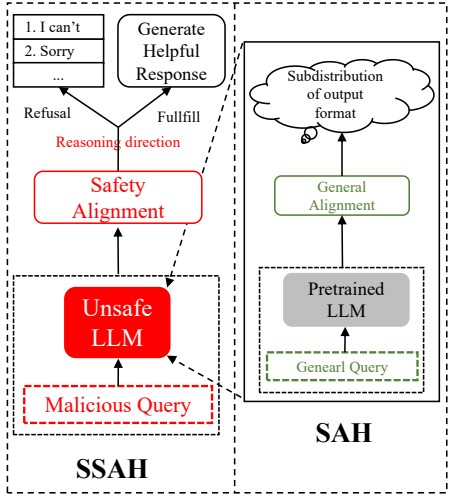

Figure 1: **Superficial (Safety Alignment (ours) v.s. Alignment) Hypotheses**

**Question 2.** *Why is safety alignment brittle, and why does it introduce an alignment tax?*

**Answer:** We propose an attribute-based approach to analyzing the alignment and fine-tuning processes, where specific attributes are assigned to each individual computational unit - primarily input channels and output neurons. **Our findings explain that the desired attributes can be achieved by repurposing units that were originally responsible for other functions.** This reallocation helps explain both the brittleness of safety mechanisms and alignment tax.

**Question 3.** *Can these issues of safety alignment be mitigated?*

**Answer:** By freezing the safety-critical components during fine-tuning and repurposing redundant units, we can effectively mitigate the brittleness and minimize the alignment tax. **We conclude that the atomic functional unit for safety in LLMs resides at the neuron level and underscores that safety alignment should not be complicated**.

## 2 RELATED WORK

**Alignment and Safety Alignment:** Alignment research in LLMs aims to ensure that models follow human instructions and align with human preferences across various tasks. Early work, such as Askell et al. (2021); Bai et al. (2022), focused on enabling LLMs to *"Follow instructions and be helpful, truthful, and harmless"* throughout the alignment process. Various alignment strategies have since been explored (Wang et al., 2024), including Supervised Fine-Tuning (SFT) (Taori et al.,

2023; Zhou et al., 2024), Reinforcement Learning with Human Feedback (RLHF) or AI Feedback (RLAIF) (Ouyang et al., 2022; Lee et al.), Instruction Tuning (Wei et al., 2021), Contrastive Learning (Rafailov et al., 2024; Xu et al., 2024), and Conditional Learning (Korbak et al., 2023). However, researchers have realized that achieving helpfulness, truthfulness, and harmlessness presents distinct challenges. More recent work has, therefore, shifted focus specifically toward the challenge of harmlessness, leading to an increasing emphasis on safety alignment—ensuring models avoid harmful outputs while maintaining utility Wei et al. (2024b); Qi et al. (2023).

**Alignment Tax and Fine-tuning Attack:** The process of aligning LLMs with human preferences often incurs an "alignment tax", where models experience degraded performance on downstream tasks due to the trade-offs required to maintain alignment (Bai et al., 2022; Ouyang et al., 2022; Lin et al., 2024; Wang et al., 2024). Additionally, fine-tuning attacks present another challenge: Yang et al. (2023); Qi et al. (2023)have shown that fine-tuning LLMs, even with benign data, can weaken safety measures. These findings highlight the inherent tension between safety alignment and utility, where improvements in one area often come at the expense of the other.

**Model Pruning:** Model pruning is a technique that reduces model size by removing redundant parameters, neurons, channels, layers, etc., which decreases storage needs and computational complexity without substantially impacting performance, namely magnitude pruning, unstructured/structured pruning, etc Frantar & Alistarh (2022); Frankle et al. (2020); Anwar et al. (2017); An et al. (2024); Li et al. (2024); Molchanov et al. (2019); Han et al. (2015); Lee et al. (2019); Renda et al. (2020). This method identifies and removes parts of the model that contribute least to its function, such as model weights with small magnitudes. In doing so, pruning extracts an efficient sub-model that runs faster on resource-constrained devices. We employ pruning as a tool to find out and delineate components to contribute to safety, utility, and both, respectively.

## 3 SUPERFICIAL SAFETY ALIGNMENT HYPOTHESIS (SSAH)

Previous research proposed Superficial Alignment Hypothesis (SAH): A model's *knowledge and capabilities* are learned almost entirely during *pretraining*, while *alignment* teaches the model *which subdistribution of formats* should be used when interacting with users (Zhou et al., 2024).

However, this claim is centered on general alignment, and directly validating the hypothesis is challenging due to the complex interplays between pretraining and alignment. When a model fails to fulfill a user's request, it can be difficult to determine whether the issue stems from the *pretraining stage* (due to lack of sufficient knowledge) or from the *alignment process* (due to misalignment in the output format). For example, when a model struggles with solving a math problem, it could either be a lack of relevant mathematical knowledge or the inability to structure its reasoning effectively. In such cases, good instruction techniques like the *Chain-of-Thought* approach can significantly enhance the quality of the model's responses (Wei et al., 2022).

**Superficial Safety Alignment Hypothesis (SSAH).** Since our focus is specifically on safety alignment, which has distinct properties compared to general alignment, we carefully define the scope of our hypothesis. A key observation here is that, for a model to be able to fulfill a malicious request, it must already possess the necessary knowledge and reasoning ability to carry out that harmful action. Based on this observation, we propose Superficial Safety Alignment Hypothesis (SSAH):

> **SSAH**: *Given an unsafe model that is capable of fulfilling users' malicious requests,* **safety alignment** *teaches the model the correct* **reasoning direction** *and a simple refusal mechanisms with reserved options.*

Reasoning direction here refers to the model's internal decision-making process when confronted with a malicious query. That is, it represents the path the model is inclined to take in such a binary classification task, whether to fulfill the harmful request or to issue a refusal. As illustrated in Fig. 1, compared with SAH which targets the general alignment, our **SSAH** focuses on the safety alignment and has the following key differences :

(1) **Knowledge and reasoning ability**: Safety alignment simplifies the problem by focusing specifically on models that already possess sufficient *knowledge* and *reasoning abilities*, as these models are capable of fulfilling malicious requests. This approach allows us to disregard other influencing factors and concentrate solely on the safety alignment process.

(2) **Refusal mechanisms with reserved fallback options**: Safety alignment generally requires the model to respond with a relatively *standardized refusal format*, which is simpler compared to general alignment, where a wider range of human preferences must be handled. The model can further simplify this purpose by effectively *embedding* multiple options of refusal response for all unsafe queries, such as "I cannot fulfill your request as it violates safety guidelines." or "I am unable to assist with that as I am an AI programmed to follow ethical standards."

(3) **Correction of reasoning direction**: Safety alignment also distinguishes itself by its specific goal of teaching the model to choose the *correct reasoning direction*, which involves either *fulfilling* or *refusing* a user's request based on whether it is safe. This process can be interpreted as a simple binary classification task.

Our **SSAH** can even provide insight into jailbreak attacks, where models often struggle to defend against adversarial inputs. In such cases, attackers typically use manipulative tokens to bypass the model's safety mechanisms, indicating that the current alignment method can only hold the correct reasoning direction in limited generated tokens. However, a potential solution inspired by SSAH suggests that if safety alignment equips the model with the ability to consistently re-select the correct reasoning direction at each step (by re-evaluating the context of previously generated tokens before producing the next one), the model can continue to generate outputs that are both safe and helpful, even in the face of adversarial attempts.

**Challenges in Proving.** While SSAH provides a more specific focus than SAH, empirically proving it still presents significant challenges. A key issue is the infeasibility of sampling sufficient outputs to fully capture the model's distribution of responses across both *safety-aligned* and *non-safety-aligned* models. This challenge makes it hard to draw comprehensive and profound conclusions or interpret certain model behaviors solely from benchmark outputs.

However, we approach the problem from an alternative perspective: if SSAH holds, we should observe distinct and consistent differences in the reasoning direction at each step of generation between safety-aligned and non-safety-aligned models. In a safety-aligned model, the reasoning direction should consistently guide the model in rejecting harmful queries at every token generation step. In contrast, a non-safety-aligned model might exhibit reasoning patterns that lean toward fulfilling malicious requests. Rather than relying solely on surface-level benchmark evaluations, we can probe the model's reasoning direction to gain deeper insights into its internal decision-making process at each step regardless of the specific outputs produced.

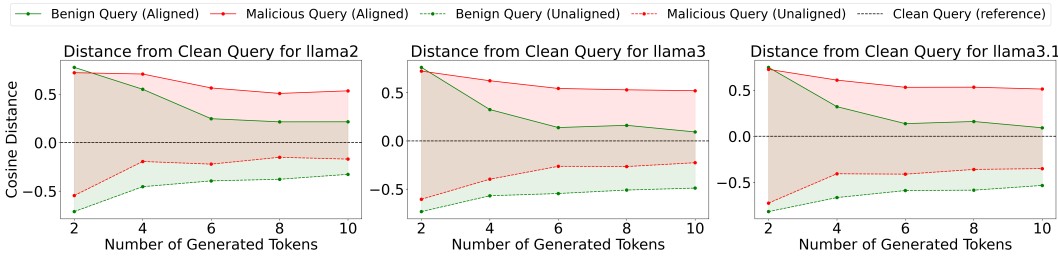

Figure 2: Probing reasoning direction on the AdvBench dataset with **Llama2-7B**, **Llama3-8B**, and **Llama3.1-8B** using cosine distance. Models were fine-tuned to ensure that aligned versions possess both general instruction-following abilities and safety guardrails, while unaligned models only have instruction-following capabilities. More results and details can be found in Appendix A.2.

**Probing Experiment.** Although we cannot directly observe the model's reasoning direction, as a turnaround, we can infer it by measuring the *distance* between hidden states in feature space when

the model is fed with queries that follow different reasoning paths. By comparing the distances in aligned and unaligned models, we gain insights into how safety alignment affects the model's reasoning direction in each generation step. Specifically, we observe that the model's behavior can be influenced by the initial tokens in its response. For instance, appending certain tokens to a malicious query can lead an *aligned model* to produce unsafe responses or prompt an *unaligned model* to generate safe ones. This indicates that *initial response tokens* can alter the model's reasoning direction. Based on this finding, we construct three types of queries to probe the model's reasoning trajectory (More details can be found in Appendix A.2):

(1) **Query**: The original malicious query (e.g., "How to make a bomb?").

(2) **Query + benign prompt tokens**: The malicious query followed by benign prompt token (e.g., "How to make a bomb? Sorry, I can't...").

(3) **Query + malicious prompt tokens**: The malicious query followed by malicious prompt tokens (e.g., "How to make a bomb? Here's how...").

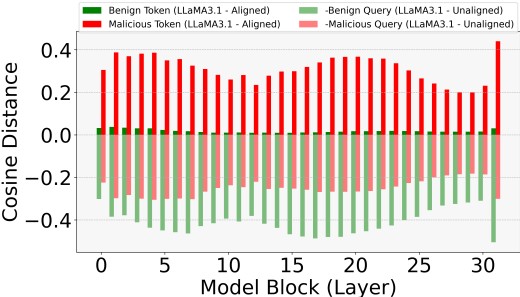

Figure 3: Cosine distance between hidden states of various queries and clean queries across all blocks of LLMs (Aligned and unaligned model definitions are the same as in Fig. 2).

**Expected Outcomes and Probe Results.** For an *aligned model*, we expect the *hidden state distances* between the **query** and **query + benign prompt token** to be shorter than those between the **query** and **query + malicious prompt token** at each generating step. In contrast, for an *unaligned model*, we anticipate the opposite: the distances between the **query** and **query + malicious prompt token** will be shorter than those between the **query** and **query + benign prompt token**. If these patterns are observed, it would indicate that safety alignment has successfully established the model's ability to choose the correct reasoning direction. This also suggests that safety alignment reshapes the model's internal decision-making process at each step of the generation, ensuring safer behavior from the very beginning of the response. As shown in Fig. 2, the probe results provide evidence that safety alignment teaches the model's correct reasoning direction as hypothesized. However, it is important to note that this evidence is necessary but not enough, as safety alignment may introduce more nuanced changes that are not fully captured by SSAH.

**Results Analysis and Discussion.** We also present the aforementioned distances and their *differences* across transformer blocks in Fig.3 and Fig.4, respectively. Our findings demonstrate that the previous conclusions are consistently upheld across all transformer blocks. Specifically, the aligned model shows larger distance differences compared to the unaligned model, suggesting that the unaligned model lacks a strong preference for safe to unsafe reasoning. In contrast, the aligned model exhibits a clear preference for safe reasoning, as reflected by the more pronounced distance differences. Moreover, we observe that in the unaligned model, the distance difference gradually increases across the earlier transformer

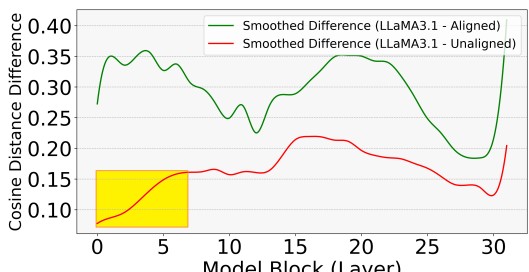

Figure 4: Absolute *differences* of cosine distance of Fig. 3 across all blocks of LLMs: *Abs( Distance( Query + Benign tokens, Query) - Distance( Query + Malicious tokens, Query))* (Aligned/unaligned model definitions are the same as in Fig. 2).

blocks 0 - 7. However, in the aligned model, the distance difference remains consistently large throughout all blocks. This indicates that the preference for safe reasoning in the aligned model is embedded not only in the later layers which typically capture higher-level features, but also in the

earlier layers. Consequently, safety alignment influences the model's internal decision-making from the initial stages of processing.

## 4 LESS IS MORE FOR SAFETY ALIGNMENT

Based on the Superficial Safety Alignment Hypothesis (SSAH), we posit that safety alignment only needs to teach the model the correct reasoning direction - either fulfilling or refusing a request - and to equip it with a standard refusal mechanism. This leads to the insight that **safety alignment can be achieved using only a small subset of critical computing units, as the task can be interpreted as a binary classification combined with a multi-selection task.**

### 4.1 IDENTIFYING SAFETY-CRITICAL COMPUTING UNITS

To verify this corollary, we designed experiments to determine the minimally essential subset of computing units in a large language model that is critical in establishing a safety guardrail. Following this line of reasoning, we hypothesize that specific attributes of LLMs can be explicitly linked to certain computing units within the model. Our experiments are designed as follows:

**Definition of Attribute Groups.** This paper categorizes the attributes of LLMs into two main properties: *utility* and *safety*. Following the above hypothesis, we first exclusively link safety or utility attributes to specific computing units. We also speculate that some units may contribute to both attributes simultaneously. Moreover, considering that many components in LLMs are redundant, we also hypothesize that certain computing units do not correlate with any attribute. Therefore, we divide the computing units of LLMs into four groups: *Exclusive Safety Units (ESU)* and *Exclusive Utility Units (EUU)*, which are linked exclusively to either safety and utility, respectively; *Complex Units (CU)*, which concurrently contribute to both safety and utility attributes; and *Redundant Units (RU)*, which are not associated with any attribute.

Table 1: Pruning results of **Llama2-7B-Chat** and **Llama3-8B-Instruct** across safety and utility benchmarks. Breakdown of model performance after pruning various categories of computing units, including ESU, EUU, and RU, demonstrating their respective contributions to safety and utility attributes. The proportion of each attribute group in the model is determined based on the degradation in utility and safety. Additional evaluation details are provided in Appendix B.1 and B.2.

| Type | wiki2 | Utility (ACC%) | | | | | | | Safety (ASR %) | | |
|------|-------|------|-------|-------|-------|--------|------|------|--------|--------|------|
| | | wino | openb | arc_c | boolq | hellas | rte | avg | w/ sys | w/o sys | avg |
| Meta-Llama-2-7B-Chat | | | | | | | | | | | |
| Dense | 6.49 | 65.5 | 32.50 | 43.5 | 79.5 | 57.0 | 71.5 | 58.3 (-0) | 3.0 | 18.0 | 10.0 (+0) |
| ESU (**1.3%**) | 6.76 | 64.0 | 34.0 | 42.5 | 78.5 | 52.0 | 70.5 | 56.9 (-1.3) | 19.0 | 84.0 | **66.0 (+56.0)** |
| EUU (13.3%) | 180.2 | 56.5 | 22.5 | 25.0 | 59.5 | 36.5 | 56.0 | **42.7 (-15.6)** | 23.0 | 48.0 | 28.3 (+18.3) |
| RU (14.8%) | 8.32 | 63.5 | 34.5 | 39.0 | 75.5 | 55.5 | 64.5 | 55.5 (-2.8) | 6.0 | 19.0 | 14.6 (+4.6) |
| Meta-Llama-3-8B-Instruct | | | | | | | | | | | |
| Dense | 7.74 | 71.5 | 34.5 | 51.0 | 80.0 | 60.0 | 70.0 | 61.2 (-0) | 2.0 | 29.0 | 15.5 (+0) |
| ESU (**1.4%**) | 9.06 | 67.0 | 30.5 | 45.0 | 82.0 | 55.5 | 65.5 | 57.6 (-3.6) | 80.0 | 93.0 | **86.5 (+71.0)** |
| EUU (6.8%) | 269.2 | 60.0 | 23.0 | 25.0 | 59.5 | 47.5 | 51.5 | **44.4 (-16.8)** | 16.0 | 24.0 | 20 (+4.5) |
| RU (6.6%) | 8.52 | 74.0 | 31.0 | 50.5 | 80.0 | 57.5 | 71.5 | 60.8 (-0.4) | 1.0 | 24.0 | 12.5 (-3.0) |

**Verfication of Attribute Group.** To verify our hypothesis that different groups of computing units contribute exclusively, collectively, or neither to safety and utility attributes, we use a model pruning mechanism. The rationale behind pruning is that removing components most closely linked to a specific attribute would significantly impact the model's performance in that area - it is a sort of ablation study. As pruning reduces the model's capacity, the most affected attributes reveal the critical components for that function.

Following Wei et al. (2024b), we construct two datasets to separately evaluate the model's performance on utility and safety. The utility dataset measures the model's functional capabilities (e.g., general reasoning, language understanding), while the safety dataset evaluates its ability to reject harmful or unethical queries. This allows us to identify the computing units most closely associated with utility and safety, respectively. Unlike previous approaches that identify safety-critical components at the weight level, we identify them at the ***neuron level***, focusing on individual neurons and channels within the model. Specifically, we use a structured pruning strategy inspired by An et al. (2024), which removes structured components of each **depth-2** module based on the variance of activation values across a target dataset. Given a depth-2 module, $f(X) = B\sigma(AX)$, which can represent either an attention module or a feedforward module, where $A$ and $B$ are weight matrices. Then, we define the importance score for each channel as follows:

$$\mathbf{I}_{:,j} = \frac{1}{N-1} \sum_{n=1}^{N} \left( X_{n,j,:}^{B} - \overline{X}_{:,j,:}^{B} \right)^2 \cdot \| \mathbf{W}_{:,j}^{B} \|_2^2 \tag{1}$$

Here, $N$ refers to the number of calibration samples, $\mathbf{W}_{:,j}^{B}$ refers to the $j$-th column of the weight matrix $B$, and $X^B$ represents the input to $B$. Based on this score, we plan to prune the input channels of $B$ and the output neurons of $A$, as channels or neurons with low activation variance across the target dataset are considered less important. More details can be found in the Appendix B.3.

In this way, we calculate the *importance score* for each individual neuron or channel, denoted as $\mathbf{I_U}$ for the utility attribute and $\mathbf{I_S}$ for the safety attribute. Initially, we prune the computing units with the smallest $\mathbf{I_U} + \mathbf{I_S}$ values to identify redundant units. Subsequently, we prune units with the largest and smallest $\mathbf{I_S} - \mathbf{I_U}$ values to identify exclusive utility and safety units, respectively. The remaining computing units are categorized as complex units. We experiment with various pruning ratios and evaluate the resulting safety and utility performance, selecting the optimal pruning ratio with minimal performance degradation in the corresponding attribute. This systematic pruning process enables us to accurately derive the roles of different units and validate our hypothesis regarding safety and utility-critical computing units.

**Ablation Study Results.** The experiment results are described in Tab. 1. We discovered that for a safety-aligned model, the computing units that are exclusively responsible for the safety attribute account for only about **1.3 - 1.4%** of the total units. Although the complex units make up a larger portion of the model, their primary role is to support the general knowledge required for both safety and utility tasks. Based on these findings, we have partially validated our hypothesis that safety alignment is relying on (**not constructed by**) a subset of safety-critical computing units.

### 4.2 WHY IS SAFETY BRITTLE?

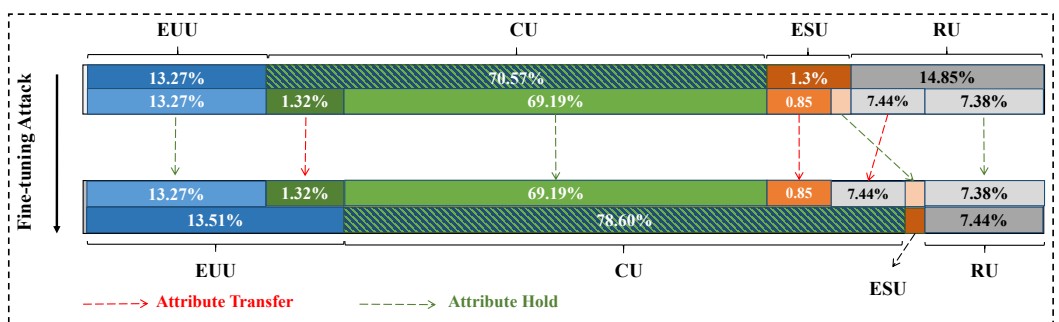

Figure 5: Attribute transfer analysis for the downstream task (Dolly Dataset) fine-tuning on **Llama2-7B-Chat**. More than half of the ESU transferred to CU, while part of the CU transferred to EUU. Although a significant portion of RU transferred to CU, this mainly contributes to utility due to the objective of the fine-tuning task. Overall, the computing units that originally contributed to safety decreased (Transfer portions less than 0.1% are excluded from this figure.)

**Attribute Transfer Analysis in the Fine-Tuning Process.**     Previous research has shown that adapting safety-aligned LLMs to new tasks can often hurt their safety performance. Therefore, it is crucial to examine how and how much the attributes of individual computing units change during this process. To investigate this, we designed an experiment where a safety-aligned LLM is fine-tuned on a downstream dataset, and we track the changes in the attributes of its computing units over time. The transfer statistics are summarized in Fig. 5, revealing several important patterns: **First**, for a safety-aligned LLM, regardless of whether it has been fine-tuned on a different task, the majority of computing units are classified as complex units. This suggests that safety and utility attributes are deeply intertwined within the aligned model. **Second**, during the fine-tuning process, a significant number of exclusive safety units and complex units are converted into exclusive utility units. This transformation indicates that fine-tuning for utility tends to shift the function of computing units away from safety, compromising the model's ability to maintain its safety guardrails.

Based on these observations, we draw the following insight: **During the task adaption of LLMs, the model often obtains the expected attribute (e.g., utility) by converting computing units that originally contributed to the other attribute (e.g., safety)**. This means that enhancing utility performance in a different task often comes at the expense of the safety performance.

Table 2: Safety performance of **Llama2-7B-Chat** and **Llama3-8B-Instruct** under Fine-Tuning attacks (Alpaca and Dolly) across various benchmarks and judge methods. We compare the measures of the initial models, fine-tuned models, and our strategies. Specifically, our strategy includes two settings: **Setting (i): freeze all ESU and the top 6% of CU**, and **Setting (ii): freeze all ESU and all CU**. Both strategies demonstrate significant mitigation of safety performance degradation. Bold indicates the best results, while the underlined mark the second-best results. Note that we doubled the training epochs for our method to ensure a fair comparison, resulting in identical or lower final training loss compared to the fine-tuned models. Additionally, due to computational limitations, we froze the first 12 transformer blocks of LLaMA3, although similar trends are observed. We also fine-tuned the model with Setting (ii) for one epoch prior to applying Setting (i). Further details are available in Appendix B.5.

| Bench | Judge | Initial | Alpaca | | | Dolly | | |
|---|---|---|---|---|---|---|---|---|
| | | | Finetuned | Fix ESU + 6%CU | Fix ESU + all CU | Finetuned | Fix ESU + 6%CU | Fix ESU + all CU |
| Meta-Llama-2-7B-Chat | | | | | | | | |
| **Adv** | keyword | 0.19% | 5.3% (+5.11%) | 2.96% (+2.77%) | **2.1% (+1.91%)** | 11.92% (+11.73%) | 3.65% (+3.46%) | **2.88% (+2.69%)** |
| | llama3-guard | 0.19% | 2.69% (+2.50%) | 1.65% (+1.46%) | **0.96% (+0.77%)** | 10.58% (+10.39%) | 2.31% (+2.12%) | **1.92% (+1.73%)** |
| **HEx-PHI** | gpt4-score | 1.05 | 1.79 (+0.74) | 1.39 (+0.34) | **1.26 (+0.21)** | 1.95 (+0.90) | 1.55 (+0.50) | **1.48 (+0.43)** |
| | gpt4-rate | 0.3% | 16.1% (+15.8%) | 7.2% (+6.9%) | **4.5% (+4.2%)** | 18.78% (+18.48%) | 10.6% (+10.3%) | **9% (+8.7%)** |
| | llama3-guard | 2.42% | 18.4% (+15.98%) | 12.12% (+9.70%) | **7.88% (+5.46%)** | 25.0% (+22.58%) | 15.0% (+12.58%) | **13.94% (+11.52%)** |
| Meta-Llama-3-8B-Instruct (Freeze 1-12 blocks%) | | | | | | | | |
| **Adv** | keyword | 1.54% | 14.24% (+12.7%) | 11.2% (+9.66%) | **10.95% (+9.41%)** | 61.15% (+59.61%) | 51.38% (+49.84%) | **40.58% (+39.04%)** |
| | llama3-guard | 1.15% | 12.88% (+11.73%) | 10.1% (+8.95%) | **9.0% (+7.85%)** | 50.58% (+49.43%) | 42.6% (+41.45%) | **28.27% (+27.12%)** |
| **HEx-PHI** | gpt4-score | 1.16 | 2.13 (+0.97) | 2.0 (+0.84) | **1.91 (+0.75)** | 2.95 (+1.79) | 2.59 (+1.43) | **2.32 (+1.16)** |
| | gpt4-rate | 3% | 23% (+20%) | 19.4% (+16.4%) | **18.7% (+15.7%)** | 37.2% (+34.2%) | 28.2% (+25.2%) | **23.6% (+20.6%)** |
| | llama3-guard | 5.75% | 33.94% (+28.19%) | 30.7% (+24.95%) | **30.3% (+24.55%)** | 60% (+54.25%) | 51.8% (+46.05%) | **42.12% (+36.37%)** |

**Freezing Safety-Critical Components to Preserve Safety Guardrails.**   Given the above insight, we propose freezing the identified safety-critical components during the fine-tuning process to prevent the unwanted attribute transfer of these units and thereby preserve safety performance. To test this hypothesis, we conduct experiments across different language models, where we freeze the safety-critical components identified through our aforementioned ablation study in the fine-tuning process. The experiment results are described in Tab. 2, and we have the following observations: **Observation 1) Brittleness of current safety mechanisms.** Adapting a fully aligned model to new tasks significantly increases the attack success rate or harmful output scores across various attack benchmarks and evaluation methods. This clearly demonstrates the brittleness of current safety mechanisms in LLMs. **Observation 2) Effectiveness of freezing safety-critical compo-**

**nents** By freezing the Exclusive Safety Units (ESU) and the top 6% of Complex Units (CU), we can significantly reduce the degradation of safety guardrails during the adaptation process across different models, tasks, benchmarks, and evaluation methods. Freezing all CU components further improves performance (The ablation experiments with varying proportions of frozen CUs are detailed in Appendix C.1). **Observation 3) Fragility of safety in newer models.** We found that the safety guardrails in LLaMA 3 are more fragile than in LLaMA 2. We speculate it because LLaMA 3 attempts to analyze the true intentions behind harmful requests, which can lead to more errors. This suggests that the community should carefully consider how models should respond to harmful requests, whether to refuse outright or engage in further interactions. **Observation 4)** Additionally, we conduct an analysis of the attribute transfer in this new setting and observe that freezing the safety-critical components successfully mitigates the conversion of safety units into utility units.

Table 3: Safety performance of **Llama2-7B-Chat** under Fine-Tuning attacks (Alpaca and Dolly) across various parameter-efficient fine-tuning setups. Results with * are from Qi et al. (2023).

| Benchmark | HEx-PHI Metric | Initial | Ours | Full | LoRA | LLaMA-Adapter | Prefix |
|---|---|---|---|---|---|---|---|
| **Alpaca** | **Harmfulness Score (1-5)** | 1.05 | **1.26 (+0.21)** | 1.79 (+0.74) | 2.18 (+1.13)* | 2.38 (+1.33)* | 2.20 (+1.15)* |
| | **High Harmfulness Rate** | 0.3% | **4.5% (+4.2%)** | 16.1% (+15.8%) | 25.2% (+24.9%)* | 26.4% (+26.1%)* | 24.8% (+24.5%)* |
| **Dolly** | **Harmfulness Score (1-5)** | 1.05 | **1.48 (+0.43)** | 1.95 (+0.9) | 2.44 (+1.39) | 2.51 (+1.46) | 2.38 (+1.33) |
| | **High Harmfulness Rate** | 0.3% | **9% (+8.7%)** | 18.78% (+18.48%) | 27.2% (+26.9%) | 27.9% (+27.6%) | 26.5% (+26.2%) |

**Comparing with Parameter-Efficient Fine-Tuning (PEFT) Approaches.** To ensure that the preservation of safety in our approach is not merely due to not updating the model itself, we also examine how safety guardrails degrade during the parameter-efficient fine-tuning of LLaMA2-7B. Specifically, we tested three PEFT methods: `LoRA`, `LLaMA-Adapter`, and `Prefix Tuning`. The experimental results are detailed in Table 2, and we found that these methods led to worse degradation of safety compared to full-model fine-tuning and even further so when compared to our approach. This indicates that the effectiveness of preserving safety in our method primarily stems from successfully identifying safety-critical components rather than simply freezing the model.

### 4.3 Free Lunch: Repurposing RUs as Alignment Budget Even Help Reduce Tax

Up to this point, we have successfully identified safety-critical components and preserved the model's safety properties while adapting LLMs to new tasks. From another perspective, an interesting question arises: **Can we assign safety or utility attributes directly to certain computing units, respectively?** If we could, this would mean that we can manage the attributes of LLMs in a fine-grained, controlled way. Building on this, the next straightforward question is whether we can **convert previously redundant units** (units that do not contribute to the utility performance of a pre-trained model) into **safety-critical units** that strengthen the safety guardrails of a safety-aligned model.

Importantly, we do not need to convert too many computing units to achieve safety alignment. Previous research has found that at least **20% of the parameters** in pretrained LLMs are **redundant** (Li et al., 2024; Ma et al., 2023; An et al., 2024). This special observation motivates us to consider the following: with such a large percentage of parameters in pre-trained LLMs available as an **alignment budget**, can we design an alignment method that reduces the **alignment tax**?

**Attribute Transfer Analysis in the Alignment Process.** Before testing the above hypothesis, we conducted an attribute transfer analysis during the alignment process to explore the underlying reasons for the alignment tax. For the pre-trained LLM, we simply categorized the attribute groups into Utility Units and Redundant Units, as the model has not yet undergone safety alignment. This is different from the categorization we used for aligned LLMs. The transfer statistics are summarized in Fig. 6, and they reveal the following key pattern: a large percentage of units that originally contributed to utility in the pre-trained model are transferred to CU and ESU in the aligned LLM. In contrast, the originally redundant units remain largely unused after the alignment process.

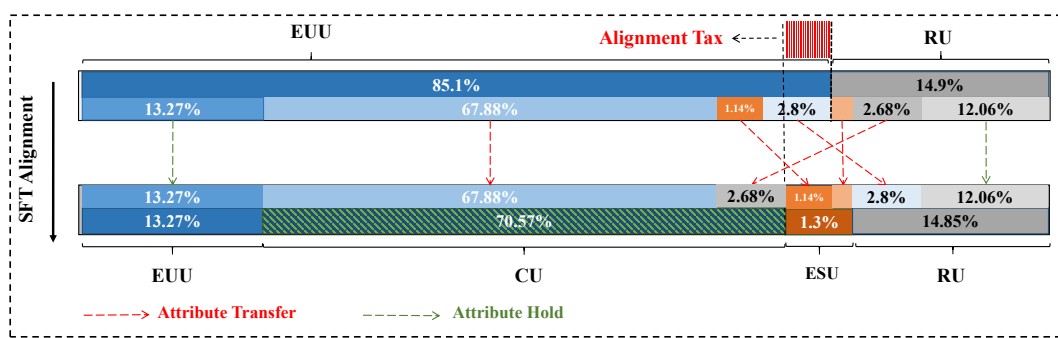

Figure 6: Attribute transfer analysis for alignment on **Llama2-7B** and **Llama2-7B-Chat**. A significant portion of computing units that originally contributed solely to utility is flipped to play a more comprehensive role after alignment. (Transfer portions less than 0.1% are excluded.)

With this observation, we are strongly motivated to verify the above question. Specifically, we use the pruning method described in Sec. 4.1 to identify the redundant units in **Llama-7B**, as there is no officially aligned model for it. Once we identify the redundant units, we freeze the updates for the rest of the model's parameters and perform fine-tuning only on these redundant units. To reduce the complexity of the experiment, we focus directly on the general alignment process instead of safety alignment, since the results should hold for the latter as a subset. The experiment results are shown in Table 4, and we successfully implemented alignment with only **20%** parameter updates and **without incurring alignment tax**, especially highlighting the mathematical performance. These findings have significant implications for the **scalability** and **efficiency** of safety alignment in future LLMs.

Table 4: Alignment results by repurposing RU in **Llama-7B**. Note that we doubled the training epochs for our method to ensure a fair comparison, resulting in identical or lower final training loss compared to the full parameters fine-tuning. Further details are available in Appendix B.6.

| LLama-7B | Type | Downstream Tasks | | | | | | | | Helpfulness (MT-bench) | |
|---|---|---|---|---|---|---|---|---|---|---|---|
| | | ARC-C | ARC-E | Hellas | Winog | Boolq | piqa | GSM8K (5 shot) | MMLU | First Turn | Second Turn |
| **Pretrained** | N/A | 44.6 | 75.2 | 76.2 | 69.7 | 75.0 | 79.2 | 9.24 | 32.20 | 1.32 | 1.02 |
| **SFT on Alpaca** | Full Parameters | 49.3 | 77.6 | 77.5 | 70.1 | 79.1 | 80.1 | **8.8 (-0.44)** | 37.8 | 2.83 | 1.47 |
| | Only RU (**20%**) | 48.9 | 77.5 | 76.6 | 70.6 | 76.9 | 80.1 | **13.4 (+4.16)** | 33.7 | 3.5 | 1.5 |

## 5    DISCUSSION, LIMITATION, AND CONCLUSION

**Discussion.** While our SSAH offers valuable insights into adversarial scenarios, such as jailbreak attacks, we do not propose a specific solution to address these issues in this work. If these issues could be resolved within the framework of our theory, the term "Superficial" in "Superficial Safety Alignment Hypothesis" may no longer be necessary. Interestingly, recent research provides some supporting evidence in this direction (Qi et al., 2024). However, it is also highly likely that advanced attacks may not be fully mitigated by relying solely on the model's internal mechanisms. A systematic, multi-layered approach, extending beyond the model itself, may be required to effectively defend against sophisticated adversarial threats.

**Limitation.** When reallocating redundant units for safety purposes, we only explored the impact of the alignment method SFT. Due to resource limitations, we have not yet tested this approach on other alignment methods like PPO or DPO.

**Conclusion.** This paper distinguishes safety alignment from the general alignment in LLMs and addresses the three key questions: How does safety alignment affect model behavior? Why are safety mechanisms brittle? and How to mitigate the safety alignment tax? By answering these questions, we were able to demonstrate that safety alignment can be a straightforward process, rather than a myth.

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

## A APPENDIX: SUPERFICIAL SAFETY ALIGNMENT HYPOTHESIS

In this section, we provide additional technical details and clarifications to supplement the experiments and findings presented in the *Superficial Safety Alignment Hypothesis (SSAH)* section. These details help ensure reproducibility and offer deeper insights into how the Superficial Alignment Hypothesis (SAH) was adapted to focus on safety-specific concerns. We also explain the methodology behind model configuration, fine-tuning, and evaluation. This appendix includes further discussion on how general instruction-following models and safety-aligned models were fine-tuned and assessed to probe their reasoning directions when faced with malicious queries, and the results of these assessments are presented in detail.

### A.1 SUPERFICIAL ALIGNMENT HYPOTHESIS IN LIMA.

The Superficial Alignment Hypothesis (SAH), as proposed to Zhou et al. (2024), fundamentally challenges the traditional assumption that a language model requires ***extensive*** fine-tuning on instruction-following on preference data to align its responses with human expectation. Instead, SAH posits that the majority of a model's knowledge and capabilities are acquired during the pretraining phase, while the subsequent alignment phase primarily functions to guide the model's output format when interacting with users. This hypothesis implies that, for many tasks, fine-tuning on a small, carefully selected set of aligned data is sufficient to achieve strong performance as long as the pretraining stage has effectively captured the necessary underlying knowledge. The key assertion of SAH is that alignment is superficial, in the sense that:

(1) Capabilities are Learned in Pretraining: During pretraining, the model acquires a vast amount of general-purpose knowledge from diverse datasets. These datasets contain implicit structures and information about language, reasoning, factual knowledge, and even ethical guidelines.

(2) Alignment Guides Output Behavior: The alignment process is not responsible for teaching the model new knowledge or capabilities. Rather, it acts as a filter that directs the model to produce acceptable formats or styles of responses based on user queries, reflecting the correct subset of its vast pretrained knowledge.

(3) For instance, when tasked with generating an informative response, the model must select a format that aligns with user expectations, such as providing clear instructions or explanations. However, the actual content of the response, e.g., factual knowledge, reasoning, and domain-specific expertise, stems from pretraining. The alignment stage merely teaches the model how to express that knowledge or when to refrain from providing information in inappropriate contexts.

**Challenges and Motivations Behind SAH.** One of the primary motivations for introducing SAH was the observation that models tend to be capable of performing certain tasks after alignment fine-tuning on a minimal dataset. This observation challenges the need for extensive fine-tuning using reinforcement learning (e.g., RLHF) or large-scale human feedback, which can be computationally too expensive and time-consuming. The authors of LiMA argue that most of the functional capabilities of a language model are already present after pretraining, and that alignment is more about conditioning the model to apply these capabilities in a user-friendly way.

The Superficial Alignment Hypothesis can also help explain phenomena where models exhibit brittleness - for example, where an LLM generates inappropriate or harmful responses in new domains or under adversarial conditions. This brittleness is attributed to the fact that alignment does not deeply alter the underlying decision-making processes of the model, but only skims the surface to adjust output behavior in specific contexts. Therefore, if an adversary finds a way to bypass these superficial alignments (e.g., via jailbreaking), the model's underlying pretrained knowledge and capabilities may still enable it to produce harmful or misaligned responses.

**Relevance to Superficial Safety Alignment Hypothesis.** While SAH deals with general alignment (i.e., ensuring that a model follows general user instructions), SSAH is specifically focused on ensuring that a model safely interacts with users, especially when faced with harmful or malicious queries. The key parallels between SAH and SSAH include:

(1) Pretrained Knowledge and Safety Concerns: Just as SAH assumes that knowledge and capabilities are largely acquired during pretraining, SSAH assumes that a model's ability to execute harmful actions (e.g., generating unsafe or unethical content) also stems from pretraining. Safety alignment, like general alignment, does not aim to teach the model new facts or capabilities, but rather to guide its reasoning pathways in a safe direction.

(2) Binary Classification in SSAH: While SAH suggests that general alignment helps models choose the correct output subdistribution, SSAH posits that safety alignment simplifies this further by focusing on a binary classification task: either fulfill a request (if safe) or refuse it (if unsafe). This simplified framing of safety alignment is consistent with the "superficial" nature of SAH, where the alignment process fine-tunes how the model behaves in response to queries, rather than altering its deep internal structures.

(3) Refusal Mechanisms and Format Control: Just as general alignment teaches models to structure their outputs in a user-friendly way, safety alignment in SSAH teaches models to issue consistent refusal mechanisms. These refusals take the form of standardized responses that indicate the model's compliance with safety guidelines, much like how general alignment might guide a model to give well-structured, polite answers to other types of questions. Importantly, this refusal mechanism makes it easier to choose the appropriate subdistribution of the output format.

The Superficial Alignment Hypothesis (SAH) as outlined in the Zhou et al. (2024) provides a theoretical framework for understanding how alignment processes operate in large language models. It suggests that alignment is largely superficial, conditioning the model on how to use its pretrained knowledge effectively. The Superficial Safety Alignment Hypothesis (SSAH) builds on this by applying similar principles to the realm of safety, simplifying the task of safety alignment to binary decisions regarding the fulfillment or refusal of unsafe requests. Both hypotheses underscore that alignment does not deeply alter the core abilities of the model, but rather adjusts the way those abilities are applied in specific contexts.

## A.2 MODEL CONFIGURATION AND TRAINING DETAILS

In our probe experiments, we explore the reasoning direction differences between unsafety-aligned models and safety-aligned models across several popular LLaMA families, including **LLaMA2**, **LLaMA3**, and **LLaMA3.1** (Fig. 7 describes more probing results on the HEx-PHI dataset). These models offer diverse pretrained knowledge and capabilities, allowing us to investigate how safety alignment affects model behavior when responding to malicious queries. To isolate the impact of *reasoning direction* when facing unsafe inputs, it is crucial to control for other confounding factors. Existing open-source instruction-following models are typically both *helpful* and *safe*, while pretrained open-source models without safety alignment are neither helpful nor safe. This dichotomy presents a challenge in disentangling the effect of general instruction-following capabilities from safety-specific behaviors.

Thus, for each LLaMA variant (LLaMA2, LLaMA3, LLaMA3.1), we fine-tuned two separate models using **Supervised Fine-Tuning (SFT)**:

(1) A **General Instruction-Following Model** that is trained to follow human instructions but without any explicit safety mechanisms. This model helps us evaluate how a model with instruction-following capabilities but without safety guardrails reacts to malicious queries.

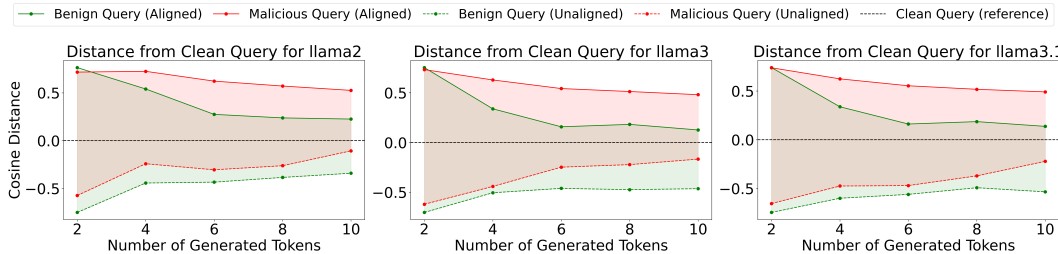

Figure 7: Probing reasoning direction on the Hex dataset with **Llama2-7B**, **Llama3-8B**, and **Llama3.1-8B** using cosine distance. Models were fine-tuned to ensure that aligned versions possess both general instruction-following abilities and safety guardrails, while unaligned models only have instruction-following capabilities. More results and details can be found in Appendix A.2.

(2) A **Safety-Aligned Model** that incorporates both general instruction-following capabilities and explicit safety mechanisms, allowing us to examine how safety alignment influences the model's reasoning direction when responding to unsafe inputs.

By comparing these two categories of models when exposed to different types of malicious queries, we can better understand how safety alignment reshapes the internal decision-making process of large language models.

**Supervised Fine-Tuning Process and Configuration.** We follow the alignment method outlined in the Zhou et al. (2024), which uses Supervised Fine-Tuning (SFT). For the general instruction-following models, we employed the **LIMA dataset**, which includes over 1000 instruction-following examples. However, we *removed 13 safety-related examples* to avoid conflating safety concerns with general instruction-following abilities. The filtering process was assisted by GPT-4, following a set of instructions specifically designed to identify and exclude safety-related tasks. For the safety-aligned models, we used the **Alert dataset**, which contains a variety of safety-critical instructions to teach the model how to respond safely to malicious queries.

For all models, we followed a consistent training configuration across the LLaMA2, LLaMA3, and LLaMA3.1 versions to ensure comparable results. The general instruction-following models were trained on the **LIMA dataset** (with safety-related data removed). The fine-tuning was performed using the following key parameters:

- **Batch size**: We set the batch size per device to 4, with gradient accumulation steps of 6 on 3 NVIDIA A6000 GPU, which gave us an effective batch size of 72.
- **Learning rate**: The learning rate was set to 1.0e-5.
- **Epochs**: The fine-tuning was conducted for 15 epochs.
- **Precision**: BF16 precision was used to optimize memory usage.
- **Optimization**: AdamW optimizer with $\beta_1 = 0.9$, $\beta_2 = 0.95$, and a weight decay of 0.1.
- **Learning rate scheduler**: We employed a linear scheduler with no warm-up steps.
- **Seed**: A random seed of 42 was used for reproducibility.

For the safety-aligned models, we fine-tuned the instruction-following models (trained on the LIMA dataset) further using the **Alert dataset**. The key fine-tuning parameters for this stage were as follows:

- **Model initialization**: We initialized the model from the previously fine-tuned general instruction-following model.
- **Batch size**: As with the general instruction model, we used a batch size of 4 with gradient accumulation steps of 6 on 3 NVIDIA A6000 GPU.

- **Learning rate**: The learning rate was set to 1.0e-5.

- **Epochs**: We fine-tuned the safety-aligned models for 9 epochs.

- **Precision**: BF16 precision was used to optimize memory usage.

- **Optimization**: The same AdamW optimizer configuration was applied.

By fine-tuning both general instruction-following and safety-aligned models with these configurations, we create a controlled environment for probing the *reasoning direction* in response to various queries. This enables us to systematically compare the behaviors of different model types under the same conditions and assess the impact of safety alignment at each generation step.

**Model Evaluation and Validation.**     To ensure that the two types of models we trained (i.e., general instruction-following models and safety-aligned models) meet the desired criteria, we conducted thorough evaluations of both their instruction-following abilities (helpfulness) and their safety performance (harmfulness).

For the instruction-following ability, we evaluated the helpfulness of the model using the **MT-Bench** benchmark (Zheng et al., 2023), which assesses the general utility and coherence of model responses across a wide range of tasks. Importantly, we use **GPT-4** as a judge to evaluate the helpfulness of the model's generated responses. Specifically, GPT-4 was used to compare the outputs of our trained models against standard task prompts in MT-Bench and assign scores based on response quality, relevance, and overall helpfulness. To evaluate the safety performance of the models, we employed two complementary benchmarks: **Adv-bench** and **HEx-PHI**. These benchmarks were chosen to comprehensively assess the models' ability to handle malicious or unsafe queries. To save space, please refer to Sec. B.5 for more details about these two datasets and the corresponding metrics.

| Model | MT-Bench (Helpfulness) | | | Adv-Bench | | HEx-PHI Bench | |
|---|---|---|---|---|---|---|---|
| | First Turn | Second Turn | Avg | Keyword | Llama3-guard | gpt-4 judge | Llama3-guard |
| **Llama-7B** | 1.32 | 1.02 | 1.18 | 100% | 96.92% | - | 94.24% |
| Llama-7B-lima-15-epochs | 2.79 | 1.14 | 1.97 | 100% | 97.5% | 3.89 | 92.37% |
| Llama-7B-alert-9-epochs | 3.3 | 1.4 | 2.35 | 3.3% | 2.3% | 1.21 | 2.6% |
| **Llama2-7B** | 1.58 | 1.0 | 1.29 | 100% | 96.73% | - | 93.0% |
| Llama2-7B-lima-15-epochs | 4.51 | 1.18 | 2.85 | 100% | 95.3% | 3.67 | 92.42% |
| Llama2-7B-alert-9-epochs | 4.67 | 1.45 | 3.06 | 2.1% | 1.09% | 1.09 | 1.15% |
| **Llama3-8B** | 2.77 | 1.01 | 1.91 | 100% | 96.35% | - | 91.82% |
| Llama3-8B-lima-15-epochs | 4.19 | 3.29 | 3.75 | 99.62% | 92.69% | 3.43 | 88.48% |
| Llama3-8B-alert-9-epochs | 4.43 | 3.55 | 3.99 | 3.1% | 1.7% | 1.16 | 1.8 % |
| **Llama3.1-8B** | 2.71 | 1.12 | 2.0 | 100% | 90% | - | 95% |
| Llama3.1-8B-lima-30-epochs | 3.81 | 3.18 | 3.50 | 99.23% | 80% | 3.71 | 87.5% |
| Llama3.1-8B-alert-9-epochs | 4.02 | 3.47 | 3.74 | 2.8% | 1.4% | 1.20 | 2.3% |

Table 5: Model Performance on MT-Bench (Instruction Following) and Safety Benchmarks (ASR)

**Results and Analysis**     Table 5 presents the evaluation results for the different models across helpfulness and safety dimensions. As expected, the **general instruction-following model** performed well in terms of helpfulness, as assessed by MT-Bench. However, it exhibited a significantly higher *Attack Success Rate* (ASR) in AdvBench and received high danger scores in the HEx-PHI benchmark. These findings confirm that while general instruction-following models can accurately follow user instructions, they fail to reject malicious or harmful requests, highlighting the absence of robust safety mechanisms. In contrast, the **safety-aligned model** maintained comparable performance in helpfulness while demonstrating significantly better safety performance. These models showed a much lower ASR in AdvBench and a lower danger score in HEx-PHI, reflecting their enhanced ability to reject adversarial and harmful inputs.

For simplicity, we refer to the general instruction-following model as the **unaligned model** and the safety-aligned model as the **aligned model**. With these two model types, we proceed to execute our probe experiments.

# B    APPENDIX: LESS IS MORE FOR SAFETY ALIGNMENT

In this section, we provide additional technical details and clarifications to supplement the experiments and findings presented in the *Less is More for Safety Alignment* section. These details will help ensure reproducibility and offer a deeper understanding of the methodology behind identifying safety-critical units, attribute transfer analysis, and the use of redundant units as an alignment budget.

## B.1    DEFINITION OF ATTRIBUTE GROUPS AND CATEGORIZATION PROCESS

As detailed in Section 4.1, we categorize the computational units (neurons and channels) of LLMs into four distinct groups: *Exclusive Safety Units (ESU)*, *Exclusive Utility Units (EUU)*, *Complex Units (CU)*, and *Redundant Units (RU)*. Exclusive Safety Units are primarily responsible for safety-related behavior, such as refusal mechanisms and detecting unsafe requests. Exclusive Utility Units are dedicated to general task performance, including natural language understanding, reasoning, and task-specific knowledge retrieval. Complex Units contribute to both safety and utility, as these attributes are intertwined at a higher level of abstraction. Finally, Redundant Units are not significantly involved in either safety or utility and are often characterized by low activation variance across tasks.

To systematically assign computing units to these groups, we employ a structured pruning strategy based on the variance of activation values. Specifically, we calculate the variance of activations across a target dataset for each neuron or channel. Neurons with higher variance contribute more significantly to the model's performance on a given task, while neurons with low activation variance are considered redundant and can be pruned. We define two separate importance scores for each unit—$I_U$ for utility-related tasks and $I_S$ for safety-related tasks. Units with extreme values in either dimension are considered Exclusive Units (either ESU or EUU), while units with significant contributions to both dimensions are classified as Complex Units (CU).

**Datasets Used for Computing $I_U$ and $I_S$.** To identify safety-critical regions in the model, we follow Wei et al. (2024b) to prepare two types of datasets: **safety dataset**, for attributing safety-related behaviors, and **utility dataset**, for attributing utility-related behaviors. Each dataset is structured in a (prompt, response) format. Specifically, the safety dataset is compiled using harmful instructions from *AdvBench* (Zou et al., 2023a). We also divide *AdvBench* into *AdvBench$_{eval}$* (100 instructions for evaluation) and *AdvBench$_{attr}$* (420 instructions for attribution). We prompt *Llama2-7B-chat* with *AdvBench$_{attr}$*, collecting responses that refrain from following harmful instructions. For the utility dataset, we filter out safety-related (prompt, response) pairs using sensitive phrase matching (Qi et al., 2023) from *Alpaca-Cleaned*, a refined version of the *Alpaca* dataset (Taori et al., 2023).

By performing structured pruning at various ratios and evaluating the impact on both utility and safety performance, we can accurately categorize the model's computing units. The pruning process involves removing the least critical units and measuring performance degradation, ensuring that our attribution of units is aligned with their actual contribution to model behavior.

## B.2    EVALUATING THE IMPACT ON BOTH UTILITY AND SAFETY PERFORMANCE

To evaluate the impact of pruning on both utility and safety performance, we measure the model's performance using established benchmarks for both attributes. Our approach closely follows the methods used by Sun et al. (2023); Wei et al. (2024b); Zou et al. (2023b), with adaptations to focus on the specific aspects of utility and safety in the context of safety alignment.

**Measuring Utility**. We evaluate the model's utility by measuring its average zero-shot accuracy across six common tasks from EleutherAI's LM Harness (Gao et al., 2021): *BoolQ* (Clark et al., 2019), *RTE* (Wang, 2018), *HellaSwag* (Zellers et al., 2019), *WinoGrande* (Sakaguchi et al., 2019), *ARC Challenge* (Clark et al., 2018), and *OpenBookQA* (Mihaylov et al., 2018). These tasks were chosen to reflect a broad range of general reasoning and language understanding capabilities.

**Measuring Safety**. We measure the model's safety by evaluating its *attack success rate (ASR)* in response to harmful instructions. Specifically, we prompt the model using *AdvBench_{eval}*, which consists of 100 harmful prompts, and collect the model's responses. We consider an attack as successful if the model's response lacks key patterns indicative of instruction rejection. The ASR is then computed as the ratio of successfully attacked prompts to the total number of prompts evaluated. Our safety evaluation includes two use cases:

1. **ASR_{Vanilla}**: This metric reflects the model's response to harmful instructions under standard, non-malicious conditions. We calculate this both with and without the system-level instructions that define the model's behavior and constraints.

2. **ASR_{Adv-Suffix}**: In this setting, the attacker optimizes adversarial suffixes to bypass the model's safety guardrails (Zou et al., 2023b). This setting allows us to test the model's resilience to manipulative inputs that are designed to mislead the model into following harmful instructions.

The **ASR_{Vanilla}** metric gives insight into the model's safety under normal operating conditions, while **ASR_{Adv-Suffix}** helps evaluate its robustness against more sophisticated attacks. By evaluating safety under both conditions, we gain a comprehensive understanding of the model's safety performance. These metrics provide a solid basis for assessing the safety and utility trade-offs during the structured pruning process, ensuring that the model maintains high utility while minimizing the risk of harmful outputs. Detailed results are provided in Table 2.

### B.3 MODEL PRUNING DETAILS AND STRUCTURED COMPONENTS

To implement structured pruning, we follow the method proposed by Li et al. (2024); An et al. (2024). The pruning process targets specific structured components (neurons or channels) within the **depth-2 modules** of the transformer architecture, which includes both attention and feedforward layers. A depth-2 module is represented as $f(X) = B\sigma(AX)$, where $A$ and $B$ are weight matrices. This paper focuses on the inner channel pruning (please refer to Fig. 1 in Li et al. (2024)): pruning the input channels of matrix $B$ and the output neurons of matrix $A$. This allows us to directly reduce the number of active channels and neurons in both the feedforward and attention mechanisms, ensuring that less important components (those with low variance) are removed.

We calculate the **importance score (I)** for each channel or neuron by measuring the activation variance across a target dataset, which is described in equation 1. For each module, channels and neurons with the least activation variance are pruned, as they are considered less critical for either utility or safety-related tasks.

In addition, to ensure consistency across layers and modules with differing scales, we apply a standardization process to the computed importance scores. Following the methodology outlined in An et al. (2024), the importance score for each channel or neuron is normalized to account for the variation in metrics across different layers and modules. The standardized importance score $\hat{I}_{:,j}^{\ell}$ for a given layer $\ell$ and channel/neuron $j$ is computed as follows:

$$\hat{I}_{:,j}^{\ell} = \frac{I_{:,j}^{\ell} - \mathbb{E}[I_{:,j}^{\ell}]}{\sqrt{\mathbb{E}[(I_{:,j}^{\ell} - \mathbb{E}[I_{:,j}^{\ell}])^2]}}.$$

Here, $I_{:,j}^{\ell}$ represents the raw importance score for the $j$-th channel or neuron in layer $\ell$, while $\mathbb{E}[I_{:,j}^{\ell}]$ represents the expected value (or mean) of the importance scores in that layer. The standard deviation is given by the square root of the variance of these scores. This standardization ensures that the

importance scores are comparable across different layers and modules, which may otherwise have widely varying metric magnitudes.

This structured pruning approach, combined with importance score normalization, ensures that the pruned units correspond to meaningful portions of the model, and the results from various pruning ratios provide insight into the essential number of units required for maintaining safety and utility performance. Although our method for identifying safety-critical components shares similarities with the design proposed by Wei et al. (2024b), we utilize a different functional structure. **Crucially, our approach has been shown to maintain safety performance under fine-tuning attacks, a result that previous methods were unable to achieve.**

### B.4 ATTRIBUTE TRANSFER DURING FINE-TUNING

In our fine-tuning experiments described in Sec. 4.2, we track the attribute transfer of individual units during the adaptation of safety-aligned models to new tasks. The process involves categorizing the computing units into ESU, EUU, CU, and RU based on their behavior in the original, safety-aligned model before and after fine-tuning. As fine-tuning progresses, we measure how many units initially classified as ESU or CU are converted into EUU or RU. This is done by re-evaluating the importance scores $I_S$ and $I_U$ for each unit after every few epochs of training.

The key insight is that when units critical to safety (ESU or CU) are re-purposed for utility tasks (becoming EUU), the model's safety performance degrades. This transformation is tracked in the *attribute transfer statistics*, which are visualized in Fig. 5 of the main text. The attribute transfer analysis highlights the brittleness of current safety mechanisms: when safety-aligned models are fine-tuned on new tasks, many safety-critical components lose their original function, compromising the safety guardrails of the model.

### B.5 EXPERIMENTAL SETUP FOR FREEZING SAFETY-CRITICAL COMPONENTS

To mitigate the safety performance degradation caused by fine-tuning, we experiment with freezing the safety-critical components identified through pruning. After categorizing the units into ESU, EUU, CU, and RU, we freeze the Exclusive Safety Units (ESU) and the top 6% of Complex Units (CU) during the fine-tuning process. This ensures that these units retain a large part of their original function and are not re-purposed for utility tasks. The rest of the model is fine-tuned as usual on new tasks, allowing the non-safety-critical components to adapt to the task while keeping the safety-critical components unchanged.

**Fine-Tuning Attack Datasets**. For fine-tuning attack experiments, we use two popular instruction-following datasets: *Alpaca* and *Dolly*. The Alpaca dataset (Taori et al., 2023) is a publicly available dataset created using GPT-3.5, and it contains 52,000 instruction-following samples across a variety of tasks. It has been widely used for instruction tuning due to its diversity in queries. The Dolly dataset (Conover et al., 2023) is another widely adopted dataset for instruction-tuning, created by Databricks, which contains high-quality examples designed to improve the model's capability to follow instructions, based on their open-source Dolly model. Both datasets allow us to effectively assess how fine-tuning for general instruction-following can impact the model's safety guardrails when safety-critical components are or are not frozen.

**Safety Evaluation Datasets**. To evaluate safety performance, we use two distinct datasets: *AdvBench* and *HEx-PHI*.

- *AdvBench* (Zou et al., 2023a): AdvBench is a benchmark designed to test a model's vulnerability to adversarial instructions. It contains prompts specifically crafted to elicit unsafe or harmful outputs. We use two evaluation methods with AdvBench. First, we employ a keyword-matching strategy (the original evaluation method) to detect whether the model produces unsafe outputs. Second, we introduce a new evaluation method using *Llama3-Guard* Inan et al. (2023), where we treat the Llama3 model as a safety arbiter (or "judge") to assess the safety of the out-

puts. The final evaluation metric is the *Attack Success Rate (ASR)*, with higher ASR indicating a more dangerous model.

- *HEx-PHI* (Qi et al., 2023): HEx-PHI is a dataset curated to evaluate model behavior on a range of ethically and safety-critical tasks. We follow prior work in using GPT-4 as a safety judge Achiam et al. (2023), where GPT-4 assigns a *dangerous score* from 1 to 5, with 1 being the least dangerous and 5 being the most dangerous. In addition to reporting the average danger score, we also compute the proportion of responses receiving the highest danger score (5). To further enhance the evaluation, we also introduce *Llama3-Guard* as an additional judge Inan et al. (2023), and compute the ASR based on its judgments, allowing for a comparison between human-aligned safety evaluations (via GPT-4) and model-aligned safety evaluations (via Llama3-Guard).

**Evaluation Metrics**. The primary metrics used in safety evaluation are the *Attack Success Rate (ASR)* and the *Dangerous Score*. For AdvBench, we compute ASR using both the keyword-matching method and the judgments from Llama3-Guard, with a higher ASR indicating that the model is more vulnerable to adversarial attacks. For the HEx-PHI dataset, we compute both the average dangerous score and the proportion of highly dangerous responses (score of 5) as evaluated by GPT-4. Additionally, we also calculate the ASR on HEx-PHI using Llama3-Guard to allow for a model-centric safety evaluation. These metrics provide a comprehensive understanding of how freezing safety-critical components impacts both general safety and the model's robustness to adversarial inputs. We evaluate the performance of these models across safety and utility benchmarks, comparing them to models that undergo full fine-tuning (with no frozen components). As reported in Table 2, freezing safety-critical components significantly preserves the model's safety guardrails, even after fine-tuning.

### B.6 DETAILS ON REDUNDANT UNITS AND ALIGNMENT BUDGET

In Section 4.3, we explore the possibility of repurposing redundant units (RU) as part of an alignment budget to minimize the alignment tax. The core idea is that pre-trained LLMs contain a large percentage of parameters that do not contribute significantly to task performance, as noted by Sun et al. (2023) and Ma et al. (2023). These redundant units can be re-purposed to improve safety alignment without sacrificing utility performance.

We identify redundant units using the same variance-based pruning method described in Section 4.1. Specifically, we compute an importance score for each neuron and channel based on the variance of activations across the Alpaca dataset (We remove safety-related samples from the original version). Once the redundant units are identified, we freeze the remaining parts of the model and fine-tune only these redundant units during the alignment process. By carefully adjusting the proportion of redundant units re-purposed, we aim to achieve alignment without incurring the alignment tax—i.e., without sacrificing utility performance. This selective fine-tuning approach significantly reduces the computational burden compared to full model fine-tuning while maintaining high task performance.

**Evaluation Benchmarks.** To evaluate the effectiveness of repurposing redundant units, we assess the model's performance on both helpfulness (MT-bench) and accuracy (downstream tasks). Our evaluations consist of two main benchmarks:

**Downstream Tasks**. As shown in Table 4, we evaluate the model's performance across a variety of tasks, including:

- **ARC-Challenge (ARC-C)** and **ARC-Easy (ARC-E)** (Clark et al., 2018): These tasks test the model's ability to answer science questions, which require a combination of factual knowledge and reasoning.

- **HellaSwag** (Zellers et al., 2019): A commonsense reasoning task requiring the model to predict the next logical action in a situation.

- **WinoGrande** (Sakaguchi et al., 2019): A commonsense reasoning task based on resolving pronoun ambiguity.

- **BoolQ** (Clark et al., 2019): A binary question-answering task that assesses the model's factual understanding.

- **PiQA** (Bisk et al., 2020): A physical commonsense reasoning task where the model must determine the most feasible solution to a given scenario.

- **GSM8K (5-shot)** (Cobbe et al., 2021): A math word problem dataset that evaluates the model's arithmetic and reasoning skills, evaluated in a 5-shot setting.

- **MMLU** (Hendrycks et al., 2020): The Massive Multitask Language Understanding benchmark tests the model's knowledge across various domains.

The results in Table 4 show that fine-tuning only on redundant units (20%) achieves performance comparable to full parameter tuning across most tasks. Notably, the model shows a significant improvement in the **GSM8K** task, suggesting that repurposing redundant units can even lead to enhanced performance on certain reasoning tasks. The minimal difference in performance between full parameter fine-tuning and redundant unit fine-tuning indicates that our approach effectively mitigates the alignment tax, preserving the model's utility capabilities.

**Helpfulness and Interaction.** Additionally, we evaluate the model's helpfulness using the **MT-bench** (Zheng et al., 2023), which evaluates how well the model engages in helpful and informative interactions over multiple turns of dialogue. The evaluation includes both *first turn* and *second turn* helpfulness scores, where the model is assessed for the usefulness of its responses. As shown in Table 4, fine-tuning only on redundant units leads to comparative or even better helpfulness scores, especially in the *first turn*, where we observe a more obvious increase compared to full parameter fine-tuning.

Overall, by repurposing redundant units for alignment, we manage to retain and even enhance the model's performance on key downstream tasks without incurring the typical alignment tax associated with full parameter updates. This approach demonstrates the potential for scalable and efficient safety alignment.

## B.7 PARAMETER-EFFICIENT FINE-TUNING (PEFT) COMPARISONS

In addition to full-model fine-tuning and freezing safety-critical components, we also tested various parameter-efficient fine-tuning (PEFT) methods, including LoRA (Low-Rank Adaptation), LLaMA-Adapter, and Prefix Tuning. These methods were evaluated for their ability to preserve safety guardrails during fine-tuning. However, as reported in Table 3, these methods exhibited worse degradation of safety compared to full-model fine-tuning and the component-freezing strategy. This suggests that merely updating a small portion of model parameters through PEFT methods is insufficient to maintain safety performance, especially when the safety-critical components are not explicitly protected. Below, we describe the configurations used for each method (Following Wei et al. (2021), we use officially recommended hyperparameters for each PEFT approach):

- **LoRA (Low-Rank Adaptation)** (Hu et al., 2021): LoRA introduces low-rank matrices into the attention mechanism, which are updated during fine-tuning while the original weights remain frozen. For our experiments, we used a **learning rate of** $10^{-4}$, a **batch size of 16**, and trained the model for **1 epoch** on the Alpaca dataset.

- **LLaMA-Adapter** (Zhang et al., 2023): This method adds small, trainable adapter modules between the layers of the transformer, allowing for parameter-efficient fine-tuning. The primary model weights remain untouched, and only the adapter weights are updated. We configured the LLaMA-Adapter with a **learning rate of** $10^{-2}$, a **batch size of 16**, and fine-tuned for **1 epoch** on the Alpaca dataset.

- **Prefix Tuning** (Li & Liang, 2021): In this approach, a set of continuous task-specific vectors (prefixes) are prepended to the input of each transformer layer, while the rest of the model

remains frozen. This method focuses on optimizing only the prefix parameters. In our experiments, we set the **learning rate to** $10^{-2}$, with a **batch size of 16**, and fine-tuned for **1 epoch** on the Alpaca dataset.

These supplementary details provide a more in-depth understanding of the technical methodologies and experimental designs used in the *Less is More for Safety Alignment* section. By outlining the structured pruning process, attribute transfer analysis, and redundant unit repurposing strategy, we ensure that our findings are transparent, reproducible, and grounded in sound experimental principles.

## C    APPENDIX: MORE PRESENTATIONS AND DISCUSSION

In this section, we provide additional presentations and discussions aimed at further enriching the understanding of how safety alignment impacts large language models and providing insights beyond the main experimental results

### C.1    ABLATION STUDY WITH DIFFERENT RATIOS OF SAFETY-CRITICAL COMPONENTS

To better understand if relying solely on the ESU is sufficient to preserve safety, we conducted experiments by freezing different ratios of safety-critical units, which included all ESUs and varying percentages of CUs. The results are shown in Fig. 8, and we found that freezing a higher percentage of CU components leads to improved safety preservation. However, when more than 9% of CU are frozen, the further safety benefits become leveling off.

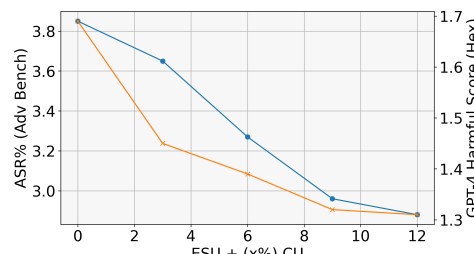

Figure 8: The degradation of safety when freezing different percent of safety-critical components on **Llama2-7B-Chat**.

### C.2    COMPARING WITH PREVIOUS WORK

While previous works have employed similar techniques (Wei et al., 2024b), such as pruning with utility or safety datasets to identify safety-critical components, our approach distinguishes itself in several key aspects:

1. **Level of Safety-Critical Component Identification**: Previous work claims to identify safety-critical components at the neuron level. However, upon closer inspection—based on their pruning metrics described in Section 2.1 and the code they provided—these components are actually identified at the weight level, as evidenced by Figure 1 in their paper. In contrast, our approach directly identifies safety-critical components at the neuron or input channel level, offering a more structured and interpretable model representation. This neuron-level identification aligns more closely with the functional model structure, making it more effective for ensuring robust safety alignment.

2. **Finer Categorization of Computational Units**: While prior methods broadly categorize computational units into two groups—those related to utility and those related to safety—we introduce a more nuanced classification with four groups: **Exclusive Safety Unit (ESU)**, **Exclusive Utility Unit (EUU)**, **Complex Unit (CU)**, and **Redundant Unit (RU)**. This more detailed categorization captures subtle yet crucial differences between various units, enabling a more precise understanding of their roles in maintaining both safety and utility.

3. **Global vs. Layer-Specific Search**: Prior work conducts a local search, focusing only on individual layers to identify safety-critical components. In contrast, our approach performs a global search across the entire model, allowing us to track the propagation of safety-critical information across multiple layers or model blocks. This global search makes our method

more flexible and comprehensive, enabling us to identify and preserve key information flows throughout the entire model or block.

4. **Robustness to Fine-Tuning**: Previous methods have struggled to maintain safety alignment even after fine-tuning on as few as 50 samples from the Alpaca dataset, despite freezing the identified safety-critical weights. Our approach, however, demonstrates far greater robustness. By freezing only **7.5%** of the identified computational units, we are able to preserve safety performance even after fine-tuning on the entire Alpaca dataset. This significant improvement in maintaining safety mechanisms while adapting to new tasks underscores the efficacy of our approach.

In conclusion, our approach offers several technical advancements over prior work, and, importantly, we are the first to achieve safety retention through such a minimal and targeted intervention. Therefore, we speculate that the atomic functional unit for safety in LLMs resides at the neuron level. This result paves the way for more efficient and scalable safety alignment strategies in future LLMs.

## C.3 ADDITIONAL FIGURE PRESENTATIONS

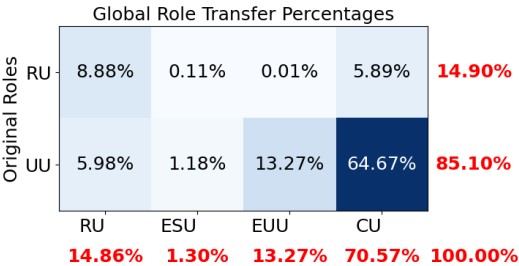

Figure 9: Global alignment process from LLaMA2-7B to LLaMA2-7B-Chat. The figure shows the conversion proportions of UU (Utility Units) and RU (Redundant Units) into different categories—RU, ESU (Exclusive Safety Units), EUU (Exclusive Utility Units), and CU (Complex Units)—during the alignment process. Each subplot illustrates the proportion of units being repurposed for different functions as safety alignment is applied, offering a global view of how the model's components are redistributed.

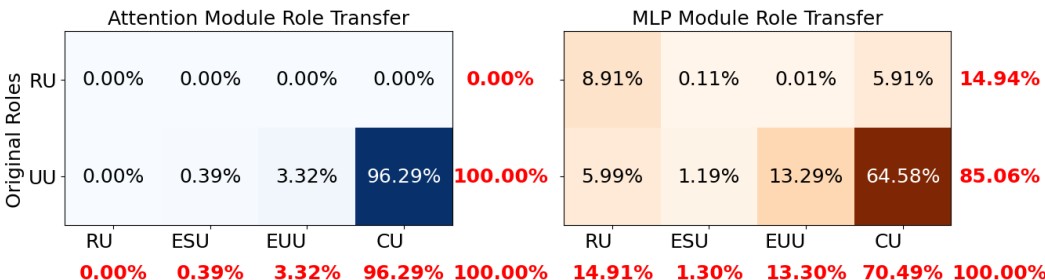

Figure 10: Alignment process for LLaMA2-7B to LLaMA2-7B-Chat, separated by Attention and MLP modules. This figure presents the conversion proportions of UU (Utility Units) and RU (Redundant Units) into RU, ESU (Exclusive Safety Units), EUU (Exclusive Utility Units), and CU (Complex Units) for the Attention (Left) and MLP (Right) modules. It provides a detailed view of how different components of the model are repurposed during safety alignment, highlighting differences between the Attention and MLP structures..

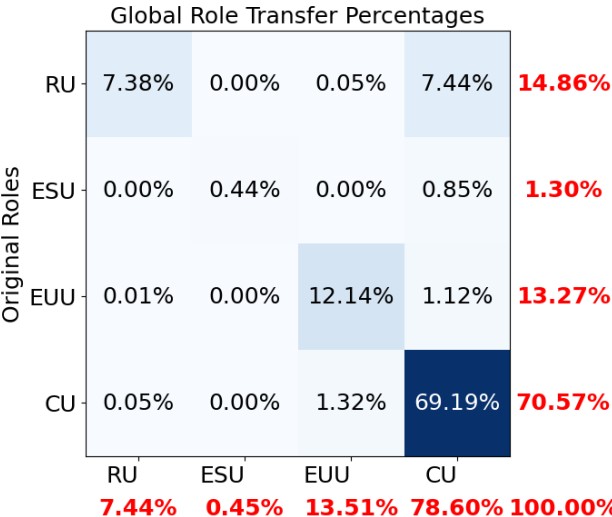

Figure 11: Role changes of Llama-7B-Chat during fine-tuning on the Dolly dataset. This figure shows the conversion proportions between the original four roles—RU (Redundant Units), ESU (Exclusive Safety Units), EUU (Exclusive Utility Units), and CU (Complex Units)—before and after fine-tuning. The 4x4 layout highlights how each of the original roles transitions into others during the fine-tuning process, providing a comprehensive view of role changes as the model adapts to the new task.

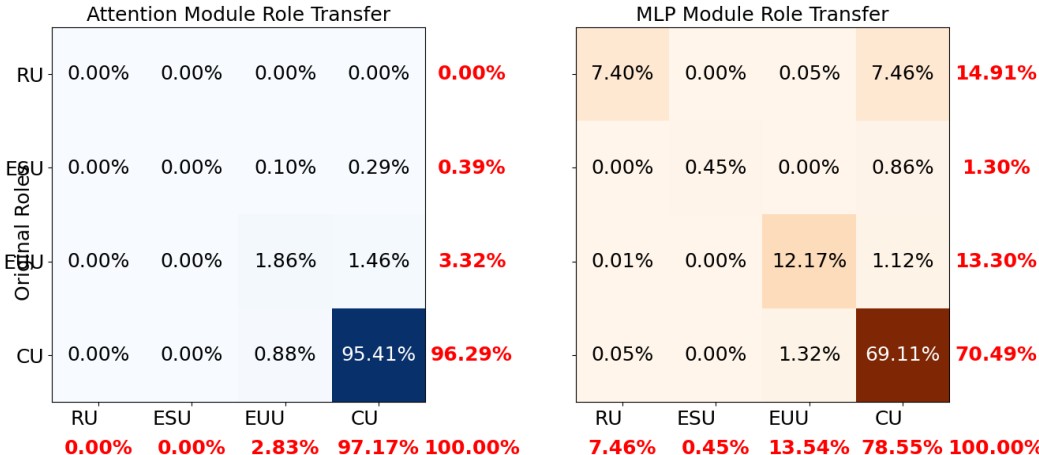

Figure 12: Role changes of Llama-7B-Chat during fine-tuning on the Dolly dataset, separated by Attention and MLP modules. This figure illustrates the conversion proportions between the original four roles—RU (Redundant Units), ESU (Exclusive Safety Units), EUU (Exclusive Utility Units), and CU (Complex Units)—before and after fine-tuning, specifically for the Attention (Left) and MLP (Right) modules.

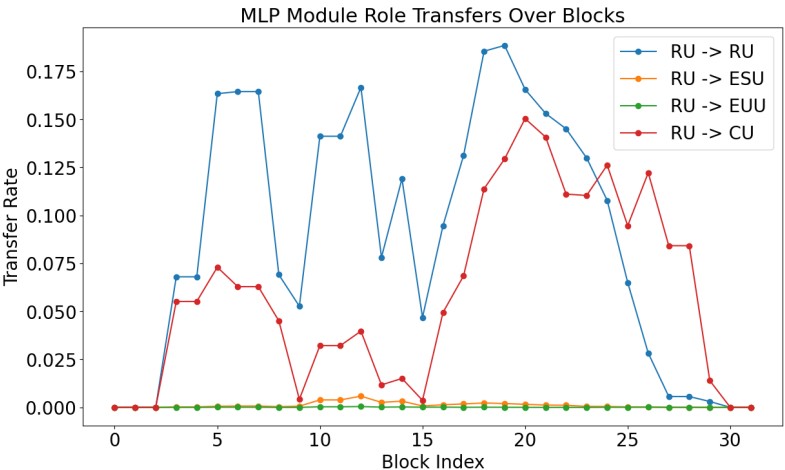

Figure 13: Conversion of RU (Redundant Units) in MLP module during the alignment process from Llama2-7B to Llama2-7B-Chat across different blocks. The majority of RU remains as RU, with a smaller portion being converted to CU and only minimal conversions into ESU and EUU. This indicates that a large percentage of redundant units remain unused and do not play any meaningful role. Additionally, RU is predominantly located in the middle blocks of the model.

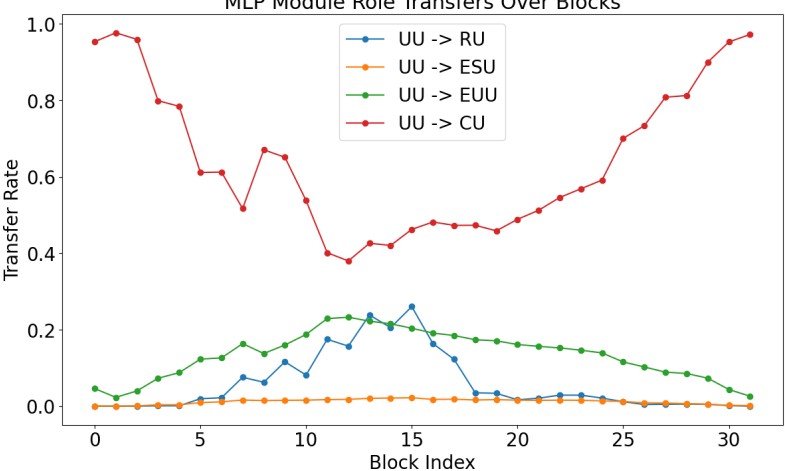

Figure 14: Conversion of UU (Utility Units) in MLP module during the alignment process from Llama2-7B to Llama2-7B-Chat across different blocks. This figure illustrates the proportion of UU being converted into other roles—CU (Complex Units), ESU (Exclusive Safety Units), EUU (Exclusive Utility Units), and RU (Redundant Units)—at various blocks of the model. Most UU are converted into CU, with a significant portion also transitioning into EUU and RU. A very small fraction are transferred to ESU. Notably, conversions to CU predominantly occur in the early and later blocks, while conversions to RU and EUU are concentrated in the middle blocks. Although transfers to ESU are minimal, they occur slightly more often in the middle blocks.

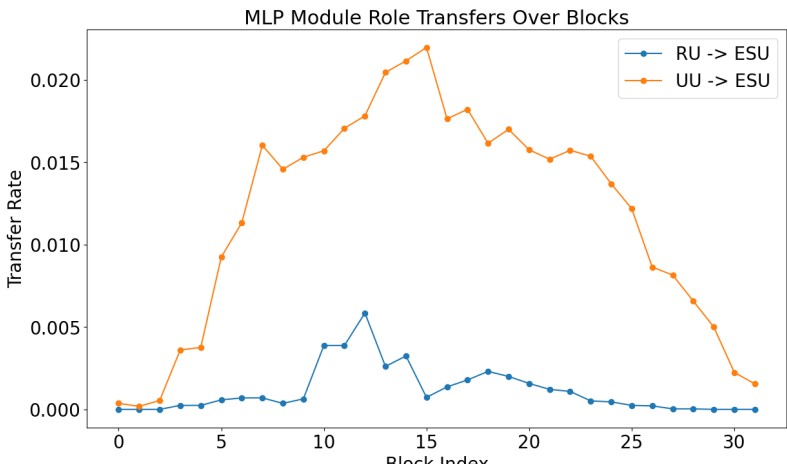

Figure 15: Conversion of units into ESU (Exclusive Safety Units) during the alignment process from Llama2-7B to Llama2-7B-Chat. Most ESU originate from UU (Utility Units) rather than RU (Redundant Units), highlighting a reduction in utility-focused units, which contributes to the alignment tax. Meanwhile, a large portion of RU remain inactive throughout the process, representing a significant inefficiency as these units fail to serve any functional role.

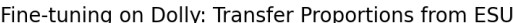

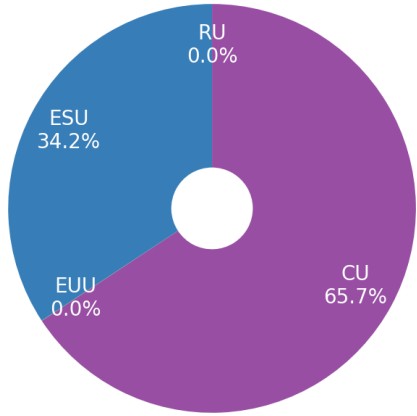

Figure 16: Conversion of ESU (Exclusive Safety Units) during the fine-tuning of Llama2-7B-Chat on the Dolly dataset. The figure shows that more than 65% of ESU are converted into other roles during the fine-tuning process, leading to a decline in the effectiveness of the safety mechanisms. This significant repurposing of safety-critical units highlights the potential risks to safety performance during task adaptation.

